# Elbow Extensor Muscles in Humans and Chimpanzees: Adaptations to Different Uses of the Upper Extremity in Hominoid Primates

**DOI:** 10.3390/ani12212987

**Published:** 2022-10-30

**Authors:** Marina de Diego, Aroa Casado, Mónica Gómez, Neus Ciurana, Patrícia Rodríguez, Yasmina Avià, Elisabeth Cuesta-Torralvo, Natividad García, Isabel San José, Mercedes Barbosa, Félix de Paz, Juan Francisco Pastor, Josep Maria Potau

**Affiliations:** 1Unit of Human Anatomy and Embryology, University of Barcelona, 08036 Barcelona, Spain; 2Institut d’Arqueologia de la Universitat de Barcelona (IAUB), Faculty of Geography and History, University of Barcelona (UB), 08001 Barcelona, Spain; 3Biological Anthropology Unit, Department of Animal Biology, Autonomous University of Barcelona, Bellaterra, 08193 Barcelona, Spain; 4Department of Anatomy and Radiology, University of Valladolid, 47005 Valladolid, Spain

**Keywords:** triceps brachii, anconeus, chimpanzee, elbow joint

## Abstract

**Simple Summary:**

Chimpanzees and humans are both species of hominoid primates that are closely related phylogenetically. One of the key differences between these two species is their use of their upper extremities. Humans use this limb mainly in manipulative tasks, while chimpanzees also use it during locomotion. In this study, we have analyzed the muscle architecture and the expression of the myosin heavy chain isoforms in the two elbow extensor muscles, the triceps brachii and the anconeus, in humans and chimpanzees, in order to find differences that could be related to the different uses of the upper extremities in these species. We have found that the triceps brachii of chimpanzees is more prepared for strength and power as an adaptation to locomotion, while the same muscle in humans is more prepared for speed and resistance to fatigue as an adaptation to manipulative activities. Our results increase the knowledge we have of the musculoskeletal system of chimpanzees and can be applied in various fields, such as comparative anatomy, evolutionary anatomy or anthropology.

**Abstract:**

The anatomical and functional characteristics of the elbow extensor muscles (triceps brachii and anconeus) have not been widely studied in non-human hominoid primates, despite their great functional importance. In the present study, we have analyzed the muscle architecture and the expression of the myosin heavy chain (MHC) isoforms in the elbow extensors in humans and chimpanzees. Our main objective was to identify differences in these muscles that could be related to the different uses of the upper extremity in the two species. In five humans and five chimpanzees, we have analyzed muscle mass (MM), muscle fascicle length (MFL), and the physiological cross-sectional area (PCSA). In addition, we have assessed the expression of the MHC isoforms by RT-PCR. We have found high MM and PCSA values and higher expression of the MHC-IIx isoform in the triceps brachii of chimpanzees, while in humans, the triceps brachii has high MFL values and a higher expression of the MHC-I and MHC-IIa isoforms. In contrast, there were no significant differences between humans and chimpanzees in any of the values for the anconeus. These findings could be related to the participation of the triceps brachii in the locomotion of chimpanzees and to the use of the upper extremity in manipulative functions in humans. The results obtained in the anconeus support its primary function as a stabilizer of the elbow joint in the two species.

## 1. Introduction

The elbow extensor muscles, the triceps brachii and the anconeus, are located in the posterior compartment of the arm and in the lateral region of the elbow, respectively [1] (Figure 1). The origin and insertion site of these two muscles have a similar anatomical structure in humans (*Homo sapiens*) and in common chimpanzees (*Pan troglodytes*)—two species that are closely related phylogenetically [2]. The triceps brachii consists of a long head, which originates in the infraglenoid tubercle of the scapula in humans and at least half of the length of the lateral border of the scapula in chimpanzees, a lateral head, which originates on the posterior aspect of the diaphysis of the humerus superior to the radial groove, and a medial head, which originates on the posterior face of the diaphysis of the humerus inferior to the radial groove [3,4]. The three heads of the triceps brachii converge and insert into the olecranon process of the ulna [1,5,6,7]. The anconeus originates in the lateral epicondyle of the humerus and inserts into the lateral aspect of the olecranon and into the upper region of the posterior aspect of the ulna [1,3,6,8,9,10]. Functionally, the triceps brachii is the main extensor of the elbow in humans [3,7,8,11], while the anconeus has a more complex function [12]. In addition to acting as an accessory extensor of the elbow [9,11,13,14], the anconeus works to stabilize the elbow [14,15,16,17,18] and is also an abductor of the ulna during pronation-supination movements [19].

Despite the close phylogenetic relationship between humans and chimpanzees [2], the main functions of the upper extremities—and hence the use of the elbow extensor muscles—differ between the two species: for locomotion in chimpanzees and for manipulative activities in humans [7]. Taxonomically, humans and chimpanzees are both hominoid primates or apes, a group that also includes bonobos, gorillas, orangutans, gibbons and siamangs [20]. These hominoid primates share common anatomical features that increase the mobility of large joint complexes in the upper extremity, such as the shoulder, elbow, and wrist [21,22,23]. These features have allowed the appearance of different forms of arboreal locomotion in non-human hominoid primates, including vertical climbing, arm suspension and brachiation [20], as well as the manipulative abilities characteristic of humans [24]. In addition, non-human African apes (gorillas, chimpanzees and bonobos) use a characteristic form of terrestrial quadrupedal locomotion called knuckle-walking [5], which represents up to 80–90% of the total locomotor repertoire in chimpanzees [25,26,27,28]. The triceps brachii of chimpanzees is highly important during the second part of the swing phase of knuckle-walking, when the elbow is extended to bring the hand forward, as well as during the support phase, when it retracts the humerus while keeping the elbow in extension to move the body over it [29]. A moderate activity of the triceps brachii of chimpanzees has also been observed during the elevation of the upper extremity over the head but not during suspensory behavior [29]. The anconeus of chimpanzees is active during the support phase of knuckle-walking—but not during the second part of the swing phase—and during the elevation of the upper extremity over the head [29].

Although the triceps brachii and the anconeus are both elbow extensors, they have clearly differentiated functions both in humans [3,6,8] and in chimpanzees [29]. Moreover, their functions are different between the two species, as they are used mainly in manipulative activities in humans [30] and mainly in arboreal locomotion and knuckle-walking in chimpanzees [29]. These functional differences between the two muscles and between the two species can translate into muscle adaptations affecting the muscle architecture and molecular composition. The study of the architecture of a given muscle can provide information on its functional characteristics, based on the calculation of muscle mass (MM), muscle fascicle length (MFL), and the physiological cross-sectional area (PCSA) [31]. MM is directly related to the ability of a given muscle to produce power [31]; MFL reflects the number of sarcomeres arranged in series and is directly proportional to the maximum rate of shortening of muscle fascicles [32]; and the PCSA reflects the number of sarcomeres arranged in parallel and is a good indicator of the ability of a muscle to generate strength [33]. The functional capabilities of a given muscle are also related to its molecular composition, mainly the expression pattern of the myosin heavy chain (MHC) isoforms [34,35,36]. In the limb muscles of humans and chimpanzees, three MHC isoforms are expressed: MHC-I, MHC-IIa and MHC-IIx [37,38,39]. The slow isoform MHC-I is mainly expressed in the slow-twitch oxidative type-I muscle fibers [37], which are characterized by a slow contraction rate, a low capacity to generate power and a high resistance to fatigue [40]. The fast isoforms MHC-IIa and MHC-IIx are mainly expressed in fast-twitch oxidative type IIa and fast-twitch glycolytic type IIx muscle fibers, respectively [37]. These fibers are characterized by a fast-twitch speed, a high capacity to generate power and a low resistance to fatigue [34], with type IIx fibers being faster, more powerful and less resistant to fatigue than type IIa fibers [40,41].

To date, few studies have compared the anatomical, structural, functional and molecular characteristics of the elbow extensors in humans and chimpanzees. Here we have studied the muscle architecture of the elbow extensors in these two species, and we have analyzed by real-time polymerase chain reaction (RT-PCR) the expression pattern of the MHC isoforms in these muscles. Our main objective was to identify significant differences that could be related to the different uses of the upper extremity in these two primate species [42]. We also sought to identify intra-species differences between the triceps brachii and the anconeus since they have slightly different functions [11]. We hypothesized that since chimpanzees use the triceps brachii mainly in arboreal and terrestrial locomotion [29], this would be reflected in higher values of the architectural parameters related to strength and power and a higher percentage of expression of the rapid MHC isoforms. We further hypothesized that since the anconeus is used in both humans and chimpanzees as a stabilizer of the elbow, differences in muscle architecture and expression of the MHC isoforms would not be so evident between the two species. Additionally, since the morphology of muscle insertion sites (enthesis) can be related to the size and activity of the muscles [43], we have also performed a quantitative analysis of the insertion sites of the triceps brachii and anconeus in the ulna to determine if structural and molecular differences between humans and chimpanzees would also translate into differences in their insertion sites. 

Our analysis of muscle architecture, expression of the MHC isoforms, and morphology of the insertion sites of the elbow extensors in humans and chimpanzees can expand the existing knowledge of this muscle group in a species of hominoid primates closely related to humans and help to better understand the functional differences between the two species. Our findings will therefore be of interest in different fields, including comparative anatomy, physical anthropology, and evolutionary anatomy.

## 2. Materials and Methods

### 2.1. Muscle and Osteological Samples

For the analyses of muscle architecture and expression of the MHC isoforms, we dissected five upper extremities of humans (five males; mean age 85.8 years [range, 81–91 years]) and five upper extremities of adult chimpanzees (four females and one male). The human individuals came from the Body Donation Service of the University of Barcelona, and the chimpanzees were provided by the Anatomy Museum of the University of Valladolid and had died in different Spanish zoos (Zoo of Madrid, Zoo of Santillana, Bioparc of Fuengirola and Primadomus of Alicante) from causes unrelated to our study. All individuals had been cryopreserved at −18 °C without chemical fixation within 24–48 h post-mortem.

For the analysis of the triceps brachii and anconeus insertion sites, 18 human ulnae were provided by the Human Anatomy and Embryology Unit of the University of Barcelona, and 19 ulnae of chimpanzees were provided by the Anatomy Museum of the University of Valladolid. The human ulnae came from ten females and eight males (mean age, 73.1 years [range, 38–97 years]), while the chimpanzee ulnae came from eight females and 11 males (all adults, as determined by epiphyseal fusion).

### 2.2. Muscle Architecture

The upper extremities of humans and chimpanzees were dissected by the same researcher (JMP). Once the adipose and connective tissue were removed, the triceps brachii and anconeus muscles were identified and anatomical data were recorded, including their origin and insertion. After the muscles were disinserted, their MM in grams was calculated using a precision scale (model Sartorius PT610 and resolution of 0.1 g). We also calculated the MM of the triceps brachii and the anconeus relative to the total body mass (%MM). For this purpose, we used an estimated body mass of 59,700 g for the male chimpanzees, 45,800 g for the female chimpanzees, and 62,500 g for the humans [44]. Photographs of the two muscles were then taken with a Canon EOS-50 camera, which made it possible to identify the architectural pattern of the muscle fascicles (Figure 2). These photographs were analyzed with the computer software ImageJ [45] to obtain the MFL in centimeters and the pennation angle (θ). These two values were measured in five different muscle fascicles: the central region of the muscle, the proximal end, the distal end, midway between the central region and the proximal end, and midway between the central region and the distal end. The average MFL and the average θ were recorded for each muscle. We then calculated the PCSA in cm^2^ for each muscle, using the formula PCSA = (MM × cos θ)/(ρ × MFL), where ρ = muscle density (1.06 g/cm^3^) [46]. Since the upper extremity muscles are generally larger in chimpanzees than in humans [47], in order to compare the two species, the absolute MFL and PCSA values were normalized based on the MM. MFL was normalized by dividing this value by MM^1/3^ (NMFL) and PCSAs were normalized by dividing this value by MM^2/3^ (NPCSA) [31]. Finally, a 1-cm^3^ sample of the central region of each muscle was obtained and cryopreserved at −18 °C in physiological saline solution for later molecular analysis.

### 2.3. Expression of MHC Isoforms

RNA was extracted from the muscle samples using the commercial RNeasy mini kit (Qiagen, Valencia, CA, USA). We used a NanoDrop 1000 Spectrophotometer to determine the concentration, purity and amount of RNA and TaqMan Reverse Transcription Reagent Kit (Applied Biosystems, Foster City, CA, USA) to synthesize cDNA. We performed reverse transcription using 330 ng of total RNA in 10 µL of RT Buffer, 22 mL of 25 mM magnesium chloride, 20 µL dNTPs, 5 µL Random Hexamers, 2 µL RNAse Inhibitor, 2.5 µL MultiScribe Reverse Transcription and RNA sample plus RNAse-free water, for a final volume of 100 µL, in the following thermal cycler conditions: 10 min 25 °C, 48 min 30 °C and 5 min 95 °C. Applied Biosystems supplied primers and probes. Primers were labeled at the 5′ end with the reporter dye molecule FAM. MYH-I (Hs00165276_m1), MYH-IIa (Hs00430042_m1) and MYH-IIx (Hs00428600_m1) genes were analyzed. In order to avoid any possible effects of post-mortem mRNA degradation, the mRNA values for each of the MHC isoforms were normalized using the reference gene ACTB [48]. The mRNA of ACTB is detectable for more than 22 days post-mortem in skeletal muscle fibers preserved at 4 °C [49], and it is one of the reference genes that is least affected by muscular degeneration [50]. We performed RT-PCR in a total volume of 20 µL in the ABI Prism 7700 Sequence Detection System (Applied Biosystems) using the following master mix conditions: 10 µL of the TaqMan Universal PCR Master Mix (Thermo Fisher Scientific, Waltham, MA, USA), 1 µL of the primers and probes, 2 µL of the cDNA and 7 µL of the RNAse-free water. We ran all samples for each gene in duplicate using the following thermal cycler conditions: 2 min 50 °C, 10 min 95 °C and 40× (15 s 95 °C, 1 min 60 °C). We used genomic DNA as negative control in each run. We captured fluorescent emission data and quantified mRNA concentrations by using the critical threshold value and 2-∆∆Ct. Finally, we calculated the percentage of expression of each MHC isoform relative to the total expression of all MHC isoforms (%MHC-I, %MHC-IIa and %MHC-IIx).

### 2.4. Muscle Insertion Sites

To obtain the area of the triceps brachii insertion site (TBIA) and the anconeus insertion site (AIA) at the ulna in mm^2^, the 18 human and 19 chimpanzee ulnae were scanned with a Next Engine 2000 Ultra HD laser scanner (NextEngine, Inc., Santa Monica, CA, USA) and the resulting triangle meshes were edited with the MeshLab 2021.05 software [51]. When we had identified the insertion sites of the triceps brachii and the anconeus in the olecranon process of the ulna, in each of the 3-D models, we manually eliminated the remaining bone surface with MeshLab to automatically calculate the TBIA and AIA values. Finally, the TBIA and AIA values were normalized by dividing these values by the length of the ulna in each individual (NTBIA and NAIA).

### 2.5. Statistical Analysis

The normality of the sample was tested using the Shapiro-Wilk test. Variables with a normal distribution were compared with the parametric T-test and variables without a normal distribution were compared with the non-parametric Mann-Whitney U test. All variables related to muscle architecture, expression of MHC isoforms, and area of muscle insertion sites were compared between humans and chimpanzees. Variables related to muscle architecture and expression of MHC isoforms were also compared between the triceps brachii and anconeus in each species.

## 3. Results

### 3.1. Muscle Architecture

The main findings on muscle architecture are summarized in Table 1. The only significant differences between humans and chimpanzees were observed in the triceps brachii. Chimpanzees had significantly higher MM (377.5 ± 122.4 g vs. 165.4 ± 39.5 g; *p* = 0.006), %MM (0.78 ± 0.26 vs. 0.26 ± 0.06; *p* = 0.003) and NPCSA (0.69 ± 0.05 vs. 0.46 ± 0.08; *p* < 0.001) values, while humans had significantly higher NMFL values (2.04 ± 0.39 vs. 1.18 ± 0.11; *p* = 0.001). No significant differences between the two species were observed in the anconeus (Figure 3). 

In humans, there were no significant differences between the triceps brachii and the anconeus in NMFL (2.04 ± 0.39 vs. 1.87 ± 0.27; *p* = 0.445) or in NPCSA (0.46 ± 0.08 vs. 0.45 ± 0.05; *p* = 0.754). In contrast, there were significant differences between the two muscles in chimpanzees: the anconeus had higher NMFL values (1.96 ± 0.29 vs. 1.18 ± 0.11; *p* = 0.001), while the triceps brachii had higher NPCSA values (0.69 ± 0.05 vs. 0.43 ± 0.06; *p* = 0.009).

### 3.2. Expression of MHC Isoforms

The main findings on the expression of MHC isoforms are summarized in Table 2. In the triceps brachii, the percentage of expression of the MHC-IIx isoform was significantly higher in chimpanzees (43.2 ± 9.8% vs. 29.9 ± 6.0%; *p* = 0.032), while the percentage of expression of the MHC-I and MHC-IIa isoforms was higher—though not significantly so—in humans (MHC-I: 33.1 ± 3.1% vs. 27.2 ± 8.6%; *p* = 0.194; MHC-IIa: 37.0 ± 6.0% vs. 29.5 ± 14.4%; *p* = 0.316). In one human and one chimpanzee, the analysis of the MHC isoforms in the anconeus was not informative, but in the remaining eight samples, the percentage of expression of all three MHC isoforms in the anconeus was similar in humans and chimpanzees (Figure 4). 

In humans, the triceps brachii had a higher percentage of expression of the MHC-IIx isoform than the anconeus (29.9 ± 6.0% vs. 19.2 ± 9.4%; *p* = 0.075), while the anconeus muscle had a higher percentage of expression of the MHC-IIa isoform (46.4 ± 10.3% vs. 37.0 ± 6.0%; P = 0.129), but neither of these differences was statistically significant. In contrast, in chimpanzees, the triceps brachii had a significantly higher percentage of expression of the MHC-IIx isoform than the anconeus (43.2 ± 9.8% vs. 20.8 ± 6.5%; *p* = 0.006). In addition, in chimpanzees, the percentage of expression of the MHC-I and MHC-IIa isoforms was higher—though not significantly so—in the anconeus than in the triceps brachii (MHC-I: 32.5 ± 4.1% vs. 27.2 ± 8.6%; *p* = 0.306; MHC-IIa: 46.7 ± 5.2% vs. 29.5 ± 14.4%; *p* = 0.060).

### 3.3. Muscle Insertion Sites

Chimpanzees had significantly longer ulnae (291.3 ± 27.4 mm vs. 246.8 ± 24.8 mm; *p* < 0.001) and larger TBIA (531.3 ± 174.1 mm^2^ vs. 343.3 ± 122.9 mm^2^; *p* = 0.001) and AIA (615.3 ± 235.9 mm^2^ vs. 415.1 ± 137.5 mm^2^; *p* = 0.007). Chimpanzees also had significantly higher NTBIA values (1.84 ± 0.62 vs. 1.38 ± 0.43; *p* = 0.013) and non-significantly higher NAIA values (2.11 ± 0.80 vs. 1.70 ± 0.60; *p* = 0.114).

## 4. Discussion

The functional differences between human and chimpanzee elbow extensors and between triceps brachii and anconeus in each of these species have a greater effect on muscle architecture than on the expression of MHC isoforms. The triceps brachii, the largest and strongest arm muscle [7], is the main elbow extensor in both humans and chimpanzees. In chimpanzees, the triceps brachii is involved in the elbow extension that occurs at the end of the swing phase of knuckle-walking [29] and in the elevation of the upper extremity during arboreal locomotion [29,52]. This function explains the higher values for absolute and relative MM (related to power) and PCSA/NPCSA (related to strength) observed in chimpanzees in the present study (Figure 3). The large size of the triceps brachii in chimpanzees [32,33,53,54] mirrors the greater need for generation of strength in the upper extremity of non-human hominoid primates [55] since it is used in arboreal and terrestrial locomotion [32]. In fact, the upper extremity in chimpanzees accounts for 16% of their total body mass, while in humans, who do not generally use the upper limb for locomotion, it accounts for only 9% of total body mass [47]. 

The larger size of the triceps brachii in chimpanzees may also be a compensation for its relatively shorter lever arm [1]. The olecranon process of the ulna, the only insertion site of the triceps brachii, is shorter in non-human hominoid primates than in humans [5,7,20,56]. This permits the rapid elbow extension that is required during the swing phase of suspensory and climbing locomotion [5,7,20]. In addition, in African apes, the short olecranon process allows the hyperextension of the elbow that is necessary to stabilize the elbow during the support phase of knuckle-walking [1]. In humans, the presence of a relatively short upper limb [5], the lower relative mass of this limb [47], and a relatively long olecranon process are related to the lower participation of this limb in locomotion [32] and its specialization in manipulating, carrying, or throwing objects [4], including in the manufacture and use of tools [7,55]. This differential use of the upper extremity in humans and chimpanzees can explain the lower values for absolute and relative MM and PCSA/NPCSA (related to power and strength) and the higher values for MFL/NMFL (related to contraction speed) in humans (Figure 3).

The functional differences between humans and chimpanzees in the triceps brachii also translate into differences in the expression patterns of the MHC isoforms (Table 2), although these differences are less striking than those in muscle architecture. Our findings on the expression of MHC isoforms at the mRNA level in the human triceps brachii (Table 2) are in line with those of other studies at the protein level using ATPase staining (37.3 ± 15.8% of type-I fibers; 35.9 ± 19.0% of type-IIa fibers; 26.6 ± 13.9% of type-IIx fibers) or SDS-PAGE (32.9 ± 6.1% of MHC-I; 49.6 ± 16.1% MHC-IIa; 17.6 ± 16.5% of MHC-IIx) [57]. The expression pattern of MHC isoforms in the triceps brachii of both humans and chimpanzees is characteristic of phasic muscles, with a higher percentage of expression of the rapid MHC-II isoforms (Table 2) [34], which have a large capacity to generate power, a fast contraction speed, and a relatively low resistance to fatigue [40]. However, a higher percentage of expression of the MHC-I and MHC-IIa isoforms (related to resistance to fatigue) was observed in humans than in chimpanzees (Figure 4) although these differences were not significant. The higher percentage of expression of these two isoforms could be related to the prolonged maintenance of flexed elbow postures in humans during object manipulation [58,59]. In contrast, chimpanzees had a significantly higher percentage of expression of the MHC-IIx isoform (related to power [40,41]) (Figure 4), which is consistent with their higher values for MM and PCSA/NPCSA. These differences between humans and chimpanzees in the expression of MHC isoforms in the triceps brachii thus fit well with the different uses that these two species give to their upper extremities [42].

The functional differences between humans and chimpanzees in the triceps brachii, which are related to the larger MM of this muscle in chimpanzees and to its greater participation in arboreal locomotion and terrestrial knuckle-walking—activities involving the generation of force—also translate into osteological differences in its insertion site. A larger muscle insertion site is related to the larger size and the greater activity of the muscle [43,60], which is in line with our finding that the insertion site of the triceps brachii was larger in chimpanzees than in humans, both in terms of the absolute (TBIA) and normalized (NTBIA) values.

The function of the anconeus is more complex than that of the triceps brachii. The anconeus has been described as a weak elbow extensor [12,18] as it contributes to only 15% of elbow extension in humans [11], while the triceps brachii is responsible for the remaining 85%. The primary function of the anconeus in humans is to ensure the posterolateral stability of the elbow joint [3,8,15] and to participate in the abduction of the ulna during pronation [9,19]. Although the function of the anconeus in non-human hominoid primates has not been widely studied, considering the anatomical similarity between the human and the chimpanzee anconeus—with a similar origin and insertion, a similar absolute and relative MM (Table 1) and a close relationship between the muscle and the elbow joint capsule—we can postulate that its elbow stabilizing function is also important in chimpanzees. In fact, in contrast with the triceps brachii, the anconeus is not activated during the elbow extension that occurs at the end of the swing phase of knuckle-walking but is activated during the support phase, when it is needed to guarantee the stability of the elbow joint [29]. The lesser participation of the anconeus in the extension of the elbow and its importance as an elbow stabilizing muscle in both humans and chimpanzees could explain the absence of significant differences observed in our study in muscle architecture and in expression of the MHC isoforms (Figure 3 and Figure 4). This functional similarity of the anconeus between humans and chimpanzees can also explain the lack of significant differences in the size of the anconeus insertion site normalized by the length of the ulna.

Finally, when the triceps brachii and the anconeus were compared within each species, the different uses of these muscles by chimpanzees in arboreal and terrestrial locomotion [29] were reflected in significant differences in muscle architecture, where NPCSA was significantly higher in the triceps brachii and NMFL was significantly higher in the anconeus. The greater involvement of the anconeus in elbow stabilization could also be seen in a higher percentage of expression of the MHC-I and MHC-IIa isoforms, related to resistance to fatigue, while the participation of the triceps brachii in locomotion as an elbow extensor was shown in a significantly higher percentage of expression of the MHC-IIx isoform. In contrast, the greater involvement of the upper limb in manipulative functions in humans [55] was reflected in an absence of significant differences in muscle architecture between the triceps brachii and the anconeus. However, the functional difference between the triceps brachii (mainly as an elbow extensor) and the anconeus (mainly as an elbow stabilizer) was reflected in a higher percentage of expression of the MHC-IIx isoform in the triceps brachii and a higher percentage of expression of the MHC-IIa isoform in the anconeus. These differences, while not significant, suggest a greater capacity in the triceps brachii to generate power, related to its extensor function, and a greater resistance to fatigue in the anconeus, related to its stabilizing function [34,40,41]. 

Our study has several limitations, including the relatively small sample size. Specimens of *Pan troglodytes* are not abundant and a sample of five individuals is a considerable number in comparative anatomy studies of primates. However, our statistical results should be viewed with caution since significance could vary with a larger sample size. Specifically, we believe that some differences between humans and chimpanzees in the expression of the MHC isoforms could well be significant if the sample size were increased. Therefore, we suggest that, if possible, future studies should expand the sample of both humans and chimpanzees and include other species of hominoid primates. Another limitation is the advanced age of the human individuals, which is a result of the source of these individuals: body donations to science tend to be older individuals. This could have affected the values of the MM, since humans suffer a loss of MM with age [61], a condition known as sarcopenia. As a result, elderly humans can have 25–35% less MM than younger individuals [62], which may have affected our findings on the differences in the absolute MM of the triceps brachii between humans and chimpanzees. For this reason, we calculated the value of the MM of the triceps brachii and the anconeus relative to the estimated total body mass (Table 1) but found similar results to those obtained with the absolute MM. In addition, the older age of our humans could have affected our results regarding the expression of the MHC isoforms, which is conditioned by age [63]. Although the results we have obtained at the mRNA level are similar to those obtained at the protein level by other authors [57], it would be useful to analyze the expression of MHC isoforms in biopsies obtained from younger patients to see if the results differ from those obtained in our analysis.

## 5. Conclusions

Our analysis of the muscle architecture and the expression of the MHC isoforms have allowed us to explore differences in the elbow extensors between humans and chimpanzees. The greater involvement of the upper extremity in locomotion in chimpanzees is reflected in higher values of the parameters related to generation of power and strength in the triceps brachii (MM, PCSA/NPCSA, %MHC-IIx), while the greater involvement of the upper limb in manipulative functions in humans is reflected in higher values of the parameters related to contraction speed and resistance to fatigue (MFL/NMFL, %MHC-I, %MHC-IIa). In contrast, the main function of the anconeus as an elbow stabilizer in the two species is reflected in similar values in the two species.

Despite the limitations regarding sample size and age of our humans, we believe that our findings can help to increase scientific knowledge of the anatomical, molecular and functional characteristics of the elbow extensors in humans and chimpanzees. This muscle group, which has not been widely studied in non-human hominoid primates, is noteworthy due to its participation in the different types of locomotion used by chimpanzees, a species closely related phylogenetically to humans. The fact we have found quantifiable differences between humans and chimpanzees in the insertion site of the triceps brachii which are related to functional differences in this muscle leads us to suggest that similar studies should be carried out in the proximal epiphysis of fossil hominin ulnae [56,64,65,66]. These analyses could shed light on morphological changes that occurred during the transition from arboreal to terrestrial locomotion and provide information on whether the fossil hominin continued to use arboreal locomotion during the time it transitioned to bipedalism. Furthermore, the fact that we have been able to study muscle architecture, MHC isoforms, and osteological characteristics provides an integrated vision of the anatomy and function of this important muscle group in two highly related species with different functional uses of the upper extremity.

## Figures and Tables

**Figure 1 animals-12-02987-f001:**
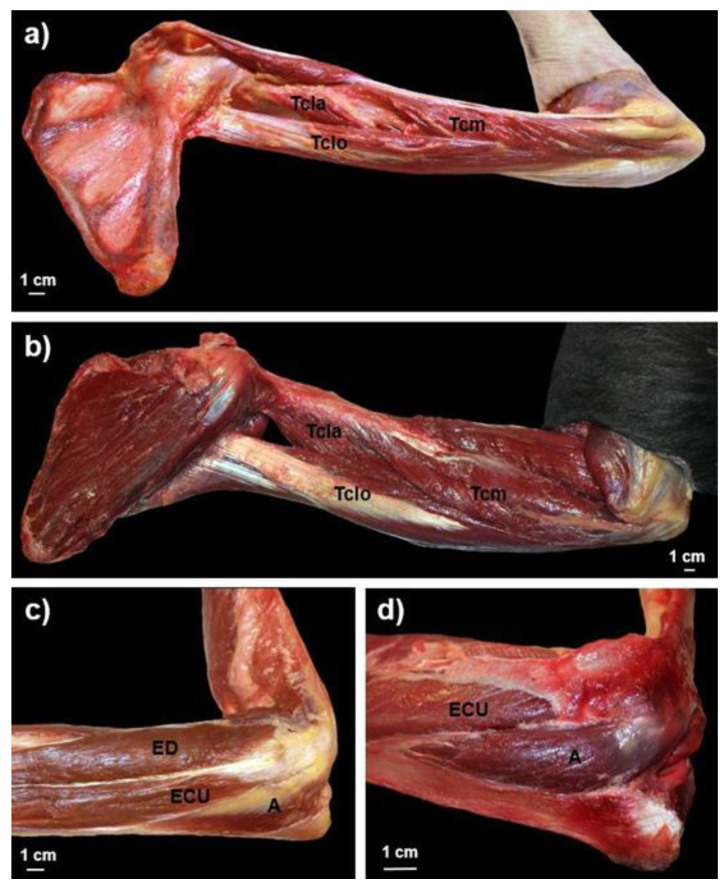
Medial view of a triceps brachii muscle dissection in (**a**) *Homo sapiens* and (**b**) *Pan troglodytes* and lateral view of an anconeus muscle dissection in (**c**) *Homo sapiens* and (**d**) *Pan troglodytes*. Tclo = Triceps brachii caput longum; Tcla = Triceps brachii caput laterale; Tcm = Triceps brachii caput mediale; A = Anconeus; ED = Extensor digitorum; ECU = Extensor carpi ulnaris.

**Figure 2 animals-12-02987-f002:**
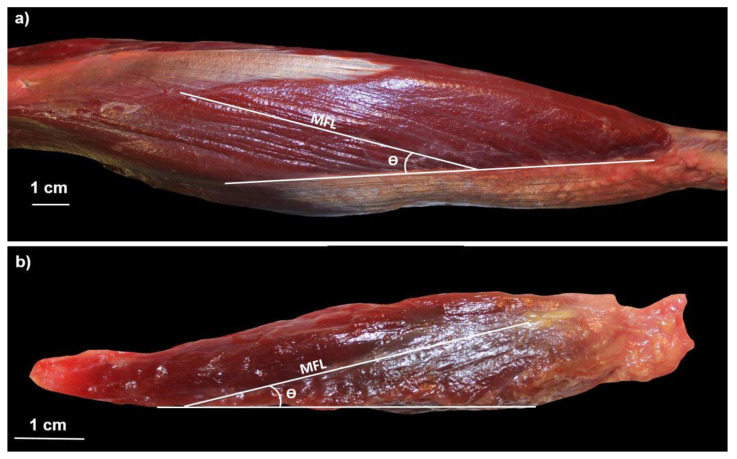
View of the (**a**) triceps brachii and (**b**) anconeus of *Pan troglodytes* displaying the architectural arrangement of their muscle fascicles. MFL = muscle fascicle length; θ = pennation angle.

**Figure 3 animals-12-02987-f003:**
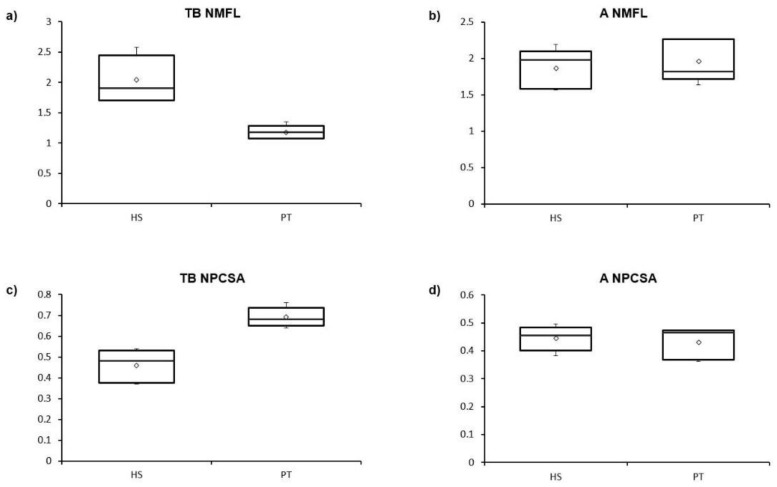
Boxplots of the differences between *Homo sapiens* (HS) and *Pan troglodytes* (PT) in NMFL and NPCSA values in the (**a**,**c**) triceps brachii (TB) and (**b**,**d**) anconeus (A).

**Figure 4 animals-12-02987-f004:**
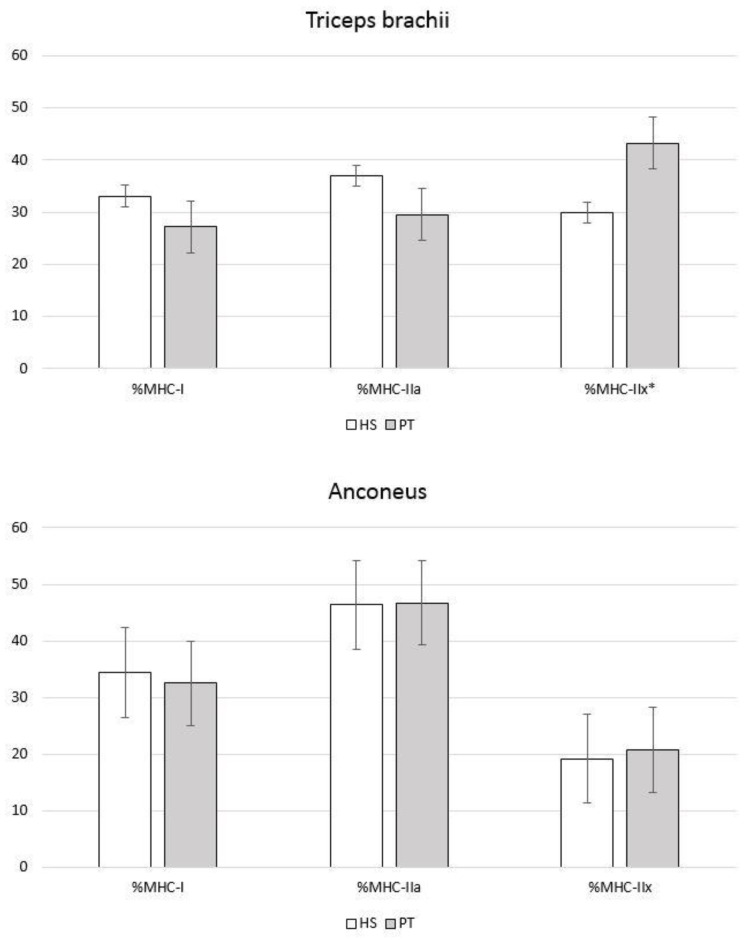
Expression of MHC isoforms in the triceps brachii and anconeus in *Homo sapiens* and *Pan troglodytes*. MHC = myosin heavy chain; HS = *Homo sapiens;* PT = *Pan troglodytes*. The data bars represent mean values, and the error bars represent the standard error. * indicates statistical significance.

**Table 1 animals-12-02987-t001:** Main findings on muscle architecture. MM = muscle mass; %MM = percentage of MM relative to total body mass; MFL = muscle fascicle length in cm; NMFL = normalized muscle fascicle length; PCSA = physiological cross-sectional area in cm^2^; NPCSA = normalized physiological cross-sectional area; HS = *Homo sapiens*; PT = *Pan troglodytes*; M = male; F = female.

SAMPLE	SEX	AGE (Years)	MM	%MM	MFL	NMFL	PCSA	NPCSA	MM	%MM	MFL	NMFL	PCSA	NPCSA
			*Triceps brachii*						*Anconeus*					
HS01	M	91	138.8	0.22	9.8	1.90	13.0	0.48	5.2	0.008	3.4	1.98	1.4	0.47
HS02	M	85	195.2	0.31	13.4	2.31	12.8	0.38	4.1	0.007	3.5	2.19	1.0	0.38
HS03	M	81	111.8	0.18	8.2	1.70	12.0	0.52	4.7	0.008	2.6	1.57	1.4	0.50
HS04	M	81	174.8	0.28	14.4	2.58	11.5	0.37	5.6	0.009	3.6	2.01	1.3	0.42
HS05	M	91	206.4	0.33	10.1	1.71	18.8	0.54	7.7	0.012	3.2	1.60	1.8	0.45
*Mean*			165.4	0.26	11.2	2.04	13.6	0.46	5.5	0.009	3.3	1.87	1.4	0.45
*SD*			39.5	0.06	2.6	0.39	2.9	0.08	1.4	0.002	0.4	0.27	0.3	0.05
PT01	F	26	521.6	1.14	9.5	1.18	45.8	0.71	2.3	0.005	2.4	1.80	0.8	0.48
PT02	F	25	399.9	0.87	8.9	1.21	36.0	0.66	5.2	0.011	3.2	1.82	1.4	0.47
PT03	M	43	426.3	0.71	8.1	1.08	38.4	0.68	8.7	0.015	3.4	1.64	2.0	0.46
PT04	F	40	189.0	0.41	6.1	1.07	25.1	0.76	0.9	0.002	2.2	2.26	0.4	0.38
PT05	F	28	350.6	0.77	9.6	1.35	31.6	0.64	7.6	0.017	4.5	2.27	1.4	0.36
*Mean*			377.5	0.78	8.4	1.18	35.4	0.69	4.9	0.010	3.1	1.96	1.2	0.43
*SD*			122.4	0.26	1.4	0.11	7.7	0.05	3.3	0.006	0.9	0.29	0.6	0.06
*p* value			0.006 *	0.003 *	0.072	0.001 *	<0.001 *	<0.001 *	0.756	0.704	0.743	0.632	0.56	0.654

* statistical significance.

**Table 2 animals-12-02987-t002:** Main findings on the expression of the MHC isoforms. MHC = myosin heavy chain; HS = *Homo sapiens*; PT = *Pan troglodytes*; M = male; F = female.

SAMPLE	SEX	AGE (Years)	%MHC-I	%MHC-IIa	%MHC-IIx	%MHC-II	%MHC-I	%MHC-IIa	%MHC-IIx	%MHC-II
			*Triceps brachii*				*Anconeus*			
HS01	M	91	36.6	28.2	35.2	63.4	27.7	50.9	21.4	72.3
HS02	M	85	31.6	43.7	24.7	68.4	27.7	52.9	19.4	72.3
HS03	M	81	36.3	41.5	22.2	63.7	NA	NA	NA	NA
HS04	M	81	30.7	35.9	33.3	69.3	39.6	31.1	29.4	60.4
HS05	M	91	30.1	35.8	34.0	69.9	42.5	50.7	6.8	57.5
*Mean*			33.1	37.0	29.9	66.9	34.4	46.4	19.2	65.6
*SD*			3.1	6.0	6.0	3.1	7.8	10.3	9.4	7.8
PT01	F	A	18.3	48.2	33.5	81.7	29.1	41.7	29.1	70.9
PT02	F	A	26.9	19.7	53.4	73.1	NA	NA	NA	NA
PT03	M	A	20.8	29.8	49.4	79.2	35.4	51.4	13.2	64.6
PT04	F	A	40.3	11.8	47.9	59.7	28.8	51.0	20.3	71.2
PT05	F	A	29.8	38.2	32.0	70.2	36.6	42.7	20.7	63.4
*Mean*			27.2	29.5	43.2	72.8	32.5	46.7	20.8	67.5
*SD*			8.6	14.4	9.8	8.6	4.1	5.2	6.5	4.1
*p* value			0.194	0.316	0.032 ***	0.194	0.678	1.000	0.789	0.678

* statistical significance.

## Data Availability

Not applicable.

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
