# Peer review of "Elbow Extensor Muscles in Humans and Chimpanzees: Adaptations to Different Uses of the Upper Extremity in Hominoid Primates"

_animals, 2022, doi:10.3390/ani12212987_

Round 1
Reviewer 1 Report
Please see attached file

Reviewer 2 Report
The manuscript must be published
Reviewer 3 Report
De Diego and colleagues’ studied muscle architecture and the expression of myosin heavy chain isoforms in two elbow extensor muscles of chimpanzees and humans. In agreement with their use, one elbow extensor muscle, the triceps brachii, emphasizes strength and power properties in chimpanzees, consistent with use during locomotion, while human extensor muscles emphasize speed and resistance to fatigue, generally seen as a response to increased manipulative activities. No detectable difference in architecture of the second elbow extensor muscle, the anconeus, was found confirming its primary function to stabilize the elbow joint in both species.
This is a well-written, convincing study from conception to data collection, data analysis and presentation as well as discussion of results that left only a few open questions of general concern the authors should address in their revision.
General points (P=page; L=line]:
1. P11L386ff. The authors describe the limitation of their human sample being composed exclusively of individuals of advanced age. The authors consider the effect of age on myosin heavy chain (MHC) isoforms. While I don’t disagree that age may affect MHC isoforms as suggested by [59], I would like the authors to also discuss potential impacts of age on MM. Potentially, if MM is inversely correlated with age, the significant difference the authors found between chimpanzee and human triceps brachii MM could be an artifact of old age. The authors should exclude this possibility.
2. The study compares anatomical and functional aspects of elbow muscles for the current state of chimpanzee and human forelimb use for locomotion (chimpanzees) and manipulation (humans). It is widely believed that one of the first traits, if not the first trait, that distinguished the human lineage from apes was bipedal locomotion. It is however also clear from the fossil record that bipedalism was initially somewhat different from what we consider full, habitual bipedalism in later hominins. Soft tissue does not preserve over time, but the authors also looked at muscle insertion sites and I am wondering if there was any indication of osteological differences on the humerus and/or radius/ulna regarding triceps brachii (caput longum, caput laterale, caput mediale) or anconeus attachment sites? Perhaps the authors can talk briefly about what would be necessary to study in a fossil hominin specimen to establish the likely use of elbow extensor muscles? While hominins were probably never knuckle-walkers, for a long time they were still using the canopy to some extent, which may well have left early hominin elbow extensor muscles (and with it its attachment sites) different from present day human elbow extensor muscles (and attachment sites)? If I were to hypothesize, I would say that early hominin forelimbs show greater similarity to elbow muscle extensors in chimpanzees than they do to modern humans.
Round 2
Reviewer 1 Report
The authors have made significant changes to the manuscript. I am happy to recommend this article for publishing.